# Revealing Pathways Associated with Feed Efficiency and Meat Quality Traits in Slow-Growing Chickens

**DOI:** 10.3390/ani11102977

**Published:** 2021-10-15

**Authors:** Chotima Poompramun, Christelle Hennequet-Antier, Kanjana Thumanu, Panpradub Sinpru, Saknarin Pengsanthia, Wittawat Molee, Amonrat Molee, Elisabeth Le Bihan-Duval, Amélie Juanchich

**Affiliations:** 1School of Animal Technology and Innovation, Institute of Agricultural Technology, Suranaree University of Technology, Nakhon Ratchasima 30000, Thailand; cp.chotima@gmail.com (C.P.); panpradub.s@g.sut.ac.th (P.S.); saknarin007@gmail.com (S.P.); wittawat@sut.ac.th (W.M.); 2INRAE, Université de Tours, BOA, 37380 Nouzilly, France; christelle.hennequet-antier@inrae.fr (C.H.-A.); elisabeth.duval@inrae.fr (E.L.B.-D.); amelie.juanchich@inrae.fr (A.J.); 3Synchrotron Light Research Institute (Public Organization), Nakhon Ratchasima 30000, Thailand; kthumanu@gmail.com

**Keywords:** Korat chicken, feed efficiency, meat quality, slow-growing chicken, transcriptome

## Abstract

**Simple Summary:**

Korat is a new chicken breed with high-protein meat, low fat, and low purine content. However, the effects of improving the breed’s feed efficiency, which would enhance production, on meat quality are unknown. Hence, understanding the genetic architecture underlying feed efficiency and meat quality traits in chicken offers new opportunities toward genetic improvement. Through a weighted gene co-expression network analysis on Korat chickens, the presented results provide new information on the molecular pathways that play important roles in FE and meat quality that could help achieve the optimum feed efficiency while maintaining meat quality in Korat chickens.

**Abstract:**

Here, molecular pathways and genes involved in the feed efficiency (FE) and thigh-meat quality of slow-growing Korat chickens were investigated. Individual feed intake values and body weights were collected weekly to the calculate feed conversion ratios (FCR) and residual feed intake. The biochemical composition and meat quality parameters were also measured. On the basis of extreme FCR values at 10 weeks of age, 9 and 12 birds from the high and the low FCR groups, respectively, were selected, and their transcriptomes were investigated using the 8 × 60 K Agilent chicken microarray. A weighted gene co-expression network analysis was performed to determine the correlations between co-expressed gene modules and FE, thigh-meat quality, or both. Groups of birds with different FE values also had different nucleotide, lipid, and protein contents in their thigh muscles. In total, 38 modules of co-expressed genes were identified, and 12 were correlated with FE and some meat quality traits. A functional analysis highlighted several enriched functions, such as biological processes, metabolic processes, nucleotide metabolism, and immune responses. Several molecular factors were involved in the interactions between FE and meat quality, including the assembly competence domain, baculoviral IAP repeat containing 5, cytochrome c oxidase assembly factor 3, and myosin light chain 9 genes.

## 1. Introduction

Poultry breeding has favored high-performance animals, but it also demands quality food resources and optimal environmental conditions. Feed efficiency (FE) is an important parameter in poultry breeding that may be assessed by the feed conversion ratio (FCR) or residual feed intake (RFI) [1,2]. The former is defined as the feed intake per unit of body weight gain, and the latter is defined as the difference between observed and predicted feed consumption. Because FE is heritable, it may be efficiently selected in meat-type chickens [3,4]. These criteria have improved over the years in fast-growing production systems through targeted selection and a reduction in slaughter age [5]. Nevertheless, FE needs to be improved in slow-growing production systems where the costs remain high. The contexts for genetic improvement between fast- and slow-growing chickens are quite different. Additionally, slow-growing chickens are meant for a high-quality market; therefore, there are high meat quality standards. A good compromise between FE and meat quality is a requirement for slow-growing chickens.

Korat (KR) is a new alternative meat-type chicken. It is a crossbreed of the Thai indigenous Leung Hang Khao chicken and a Suranaree University of Technology synthetic line. It was established for developing “Small and Micro Community Enterprise Production” to promote farming, ensure food security in communities, and preserve indigenous chicken breeds. Slow-growing KR chickens are usually slaughtered at 10 weeks of age and are consumed grilled. The meat is recognized for its high protein, low fat, and low purine content [6]. However, the low FE of KR chickens results in high production costs and a lack of competitiveness, especially when the poultry industry is confronted with variations in feed costs [7,8]. Therefore, the genetic improvement of KR chickens to increase FE is necessary. However, this change must not negatively affect the meat quality, which is the outstanding characteristic of the breed.

A breed’s FE results from the feed’s digestive and absorptive capacities as well as the metabolic use of nutrients. Several genomics studies have attempted to identify the genes or metabolic pathways involved in FE at the molecular level [9,10], the digestive tissue level, such as the intestines [11], and the effector tissue level, such as the liver [12], breast muscle [13], or fat tissue [14]. Recently, the interaction between FE and breast muscle metabolism has been investigated to better understand the mechanisms that determine the greater sensitivity of high-FE chickens to muscle defects, such as the Wooden Breast [15]. Some pathways are involved in both FE and breast muscle metabolism in meat-type chickens, such as those related to energy metabolism (i.e., the carboxylic acid metabolic process, nuclear factor- nuclear factor erythroid 2-related factor 2-mediated oxidative stress response, metabolic and biosynthetic processes, and lipid synthesis), immune processes, and transporter activities [12,14,16,17] in both FE and muscle metabolism. Additionally, a study by Yang et al. [18] revealed the importance of seven genes and three pathways related to inflammatory response, immune response, and mitochondrial function. These pathways may influence FE through reactive oxygen species production and inflammatory responses in breast muscles.

Leg muscles (thigh and drumstick), however, are preferred to breast muscles in East Asia, Mexico, India, Russia, and Morocco [19], and information is lacking on the relationship between FE and the meat quality parameters of thigh muscles in slow-growing chicken lines. Leg muscle development is related to the insulin signaling pathway, adipocytokine signaling pathway, extracellular matrix–receptor interactions, focal adhesion, and the tight junction regulation of the actin cytoskeleton [20,21]. Additionally, Ouyang et al. [22] reported the involvement of different pathways using proteomic studies of muscle contraction and oxidative phosphorylation during embryonic development in chickens. This indicated that common pathways might be important for thigh muscle development along with FE pathways, such as those for metabolic and biosynthetic processes and energy metabolism. Thus, improving the FE might affect gene expression in the thigh muscles. Furthermore, parameters, such as texture and flavor, which are important indicators of quality may be influenced by the physicochemical characteristics of the meat (such as ultimate pH) as well as its chemical composition in the lipid, fat, and nucleotide contents in particular [23]. Therefore, balancing the parameters related to FE and meat quality is needed to improve FE without degrading the meat quality.

Currently, no direct link between FE- and the thigh-meat quality-related pathways has been documented. Therefore, the aim of this study was to investigate the genes and molecular pathways involved in the FE and thigh-meat quality of slow-growing KR chickens. The results provide new information on the molecular pathways that play important roles in FE and meat quality traits in slow-growing chickens.

## 2. Materials and Methods

### 2.1. Ethics Statement and Experimental Chickens

This experiment was performed at the Suranaree University of Technology. The experimental use of chickens was reviewed and approved by the Ethics Committee on Animal Use of the Suranaree University of Technology, Nakhon Ratchasima, Thailand. The document ID is U1-02631-2559.

At hatching, 75 male slow-growing KR chickens were wing-banded and vaccinated against Marek’s disease. Thereafter, they were vaccinated according to the recommendation of the Department of Livestock Development, Thailand. Individual cages (63 cm × 125 cm × 63 cm) with the floors covered in rice hulls were used to raise the chickens individually. They were fed ad libitum using starter (21% protein), grower (19% protein), and finisher (17% protein) diets at 0–3, 4–6, and 7–10 weeks of age, respectively. A nipple automatic watering system was individually installed in each cage, and water was freely available to the birds.

### 2.2. Calculation of FCR and RFI

#### FCR Calculation

FCR, which represents the Feed Conversion Ratio, is a ratio measuring the efficiency with which the bodies of livestock convert animal feed into weight gain.

The FCR for each individual was therefore estimated based on the ratio between weight gain and feed consumption.

RFI calculation:

The residual feed intake (RFI) during week *j* was first calculated weekly as follows:*RESj* = Total feed intake for week *j* (g) − (*b_0_* + *b_1_ MMW_j_* + *b_2_ BWG_j_*)(1)
where *BWG_j_* represents the body weight gain between weeks *j* and *j −* 1 (g), *MMW_j_* represents the metabolic weight estimated from the mean body weights at weeks *j* and *j −* 1 (i.e., ${\left(\frac{BW_{j}+BW_{j-1}}{2}\right)}^{0.75}$), *b_0_* represents the intercept, and *b_1_* and *b_2_* represent the partial regression coefficients. Then, we considered the cumulative RFI from hatching to week *j* (*RFIj*) as $RFIj=\sum_{i=1}^{j}RESi$.

At 10 weeks of age, the average FCR (±SD) of the population was 2.62 (±0.35). On the basis of the FCR at 10 weeks, two groups of birds, 9 with high FCR values (HFCR; FCR from 2.99 to 3.21, average body weight 1306.67 g) and 12 with low FCR values (LFCR; FCR from 1.83 to 2.26, average body weight 1594.55 g) were selected for further molecular analyses.

Birds were slaughtered at 10 weeks of age. They were stunned by electricity, bled, scalded at 60 °C, and de-feathered. Then, the carcasses were washed and put in a cold room (4 °C). A piece of thigh muscle was immediately frozen in liquid nitrogen and was stored at −80 °C before RNA extraction. The remaining part of the thigh muscle was used to measure meat quality parameters.

### 2.3. Meat Quality Measurements

The ultimate pH was measured 24 h postmortem by directly inserting the probe of a portable pH-meter (pHCore-kit, Satorius Lab Instruments GmbH, Goettingen, Germany) into the thigh muscle after it was calibrated using buffers (pH 4.01 and 7.00) at room temperature, in accordance with recommendations.

Samples that weighed approximately 4–5 g and that were approximately 1.0 × 2.0 × 0.5 cm (width, length, and height, respectively) were wrapped in absorptive pads, placed in polyethylene bags, stored for 24 h at 4 °C, and weighed again to calculate the percentage of drip loss (DL).

The water-holding capacity (WHC) index of the meat samples was determined using a method based on that of Sakata et al. [24]. Thigh muscle samples were chopped (approximately 5 g) and weighed. Samples were placed on a nylon net and were wrapped with three pieces of filter paper (Whatman No. 4; Whatman Inc., Clifton, NJ, USA). Wrapped samples were centrifuged at 3000× *g* for 20 min (Thermo Fisher Scientific, Langenselbold, Germany) to calculate the WHC.

### 2.4. Nucleotide Content Analysis

The nucleotide contents of the thighs were individually measured in accordance with the method described by Jayasena et al. [25]. Individual thigh muscle (5 g) samples were mixed with 30 mL of 0.75 M perchloric acid, homogenized for 30 s, and centrifuged at 2000× *g* (Thermo Fisher Scientific) at 4 °C for 5 min to extract nucleic acids, which were then passed through a filter paper (No.1, Whatman International Ltd.). The filtrate (5 mL) was analyzed using HPLC (HP 1260, Agilent Technologies, Santa Clara, CA, USA). The peaks of the individual nucleotides were identified using the retention times of standards of hypoxanthine, inosine, inosine-5′-monophosphate (IMP), and adenosine-5′-monophosphate (AMP) (Sigma-Aldrich Co., St. Louis, MO, USA), and the concentrations were calculated using the area under each peak.

### 2.5. Fourier-Transform Infrared Spectroscopy (FTIR) Analysis

Fourier Transform Infrared (FTIR) was applied in this study since the technique is a highly sensitive technique that can detect very small changes in the content of the biomolecule, so we expected any minor differences that may occur in the LFCR and HFCR of the chicken samples would be detected using this method. We anticipated that the used of this method would reveal any changes in the key biomolecules of the thigh muscle that were associated with different levels of FCR.

Thigh samples were chopped and spread on aluminum foil boxes. Samples were frozen at −80 °C for 24 h and were dehydrated for 24 h in a laboratory freeze-dryer. Freeze-dried thigh samples were ground into powders.

Changes in the biomolecules profiles of the thigh samples were determined using attenuated total reflectance (ATR)-FTIR spectroscopy with a single reflection ATR sampling module coupled with a DTGS detector over the 4000–400 cm^−1^ measurement range. The measurements were performed with a spectral resolution of 4 cm^−1^ with 64 scans co-added (Bruker Optics Ltd., Ettlingen, Germany). The peak areas of integration were determined using OPUS7.2 software (Bruker Optics Ltd.). The integral areas at 3000–2800 cm^−1^, 1743 cm^−1^, 1700–1600 cm^−1^, 1600–1500 cm^−1^, 1450 and 1380 cm^−1^, 1338 cm^−1^, and 1250–900 cm^−1^ in the IR region reflect the lipid and ester carbonyl of the phospholipids, amide I, amide II, CH bending, amide III, and carbohydrates and glycogen, respectively. A spectrum from each sample group was subjected to a principal component analysis.

#### Pre-Processing of FTIR Data

In the biomolecules functional group analysis, the FTIR raw spectra of LFCR and HFCR (LFCR vs. HFCR and LRFI vs. HRFI) were measured. The data from each group consisted of 45 spectra, including technical replicates. These were reduced to three spectra per group by averaging them over the technical replicates. Spectral data were normalized, and then the integrated area for each functional group, such as C-H stretching (lipid), >C=O stretching (ester carbonyl of phospholipids), C=O stretching (amide I), C-N stretching + N-H bending coupled out of face (amide II), C-N stretching + N-H bending coupled in of face (amide III), and C-O-C, C-O dominated by ring vibrations of carbohydrates C-O-P, and P-O-P (carbohydrate and glycogen), was determined.

### 2.6. Statistical Analysis of Meat Quality Parameters

Chickens with the 9 highest and 12 lowest FE values at 10 weeks of age were selected as the LFCR and HFCR and LRFI and HRFI groups, respectively. The means of the FCR and RFI groups were analyzed using Student’s *t*-tests. The results were interpreted as statistically significant at a 5% probability level.

### 2.7. Isolation of Total RNA

Total RNA from the thigh muscles was extracted using TRIzol reagent (Invitrogen, San Diego, CA, USA). Briefly, after being mixed with TRIzol, each thigh muscle sample was transferred into a 1.5-mL microtube containing chloroform and was incubated for 5 min. Samples were centrifuged at 10,000× *g* (Thermo Fisher Scientific, Waltham, MA, USA) at 4 °C for 10 min. The pellet was precipitated using isopropanol and was washed with 75% ethanol. Each pellet was dried at 25 °C for 5 min. Then, the RNA pellet was re-suspended in 20 µL nuclease-free water.

The quality of the total RNA was assessed using a NanoDrop ND-1000 spectophotometer (Thermo Fisher Scientific, Waltham, MA, USA). The 260:280 and 260:230 ratios of all of the samples were greater than 2.0. RNA integrity numbers were assessed using RNA 6000 Nano chips (Agilent Technologies), and all of the values were greater than 8.0.

### 2.8. Microarray Analyses

Microarray analyses were performed using an 8 × 60 K Agilent Technologies array (Palo Alto, CA, USA) that contains 62,976 corresponding original probes from Platform GPL20588 at the U. S. National Center for Biotechnology Information Gene Expression Omnibus microarray database. The labeling, microarray processing, and hybridization control steps as well as the other internal control steps were performed using the @BRIDGe platform (INRA, UMR GABI, Jouy-en-Josas, France). Cyanine-3 (Cy3)-labeled cRNAs were prepared using 200 ng of total RNA and a One-Color Low Input Quick Amp labeling kit (Agilent Technologies) following the recommended protocol. For each sample, 600 ng of Cy3-labeled cRNA (specific activity > 6.0 pmol Cy3/µg of cRNA) was fragmented at 60 °C for 30 min in a reaction volume of 25 µL containing 25× Agilent Fragmentation Buffer and 10× Agilent Blocking Agent in accordance with the manufacturer’s instructions. Subsequently, 25 µL of 2× Agilent Hybridization Buffer was added to the fragmentation mixture and was hybridized to the SuperPrint G3 Custom GE 8 × 60 K (Agilent Technology, design ID: G4102A) for 17 h at 65 °C in a rotating Agilent hybridization oven (Agilent Technologies). The microarray data were submitted to the Gene Expression Omnibus microarray database under the number GSE162848.

External quality control was conducted using a principal component analysis, and pairwise correlations between arrays were determined using Pearson’s correlation coefficients. Samples and hybridizations were considered of good quality.

### 2.9. Re-Annotation of the Microarray Chip on the Galgal5 Chicken Genome

The microarray chip was re-annotated to the chicken genome, Galgal5 version. The transcriptome alignment on the Galgal5 chicken assembly (EntrezGene database) of the probes deposited on the chip was conducted using the blastn algorithm, which is available in the BLAST + suite (ncbi-blast-2.6.0+), on the cluster obtained from the Genotoul bioinfo platform (http://bioinfo.genotoul.fr/, accessed on 11 October 2021). Probes with hybridization tolerances of two mismatches and a single probe-associated gene were considered correctly annotated and were kept for further analyses.

### 2.10. Differential Expression Analysis

The R/Bioconductor package Limma version 3.36.3 [26] was used to identify the probes that were differentially expressed between the high- and low-FCR chickens. The fluorescence signals of the 41,350 expressed probes were log2 transformed and were then normalized using the median of each array. The differences in the expression between the high- and low-FCR groups were tested using a moderated-statistic in the linear model for each probe. For multiple testing, *p*-values were adjusted using the Benjamini–Hochberg method to control the false discovery rate. Differences in the expression levels between high- and low-FCR chickens were computed as the log2 transformations of the fold-changes. Probes with adjusted *p*-values ≤ 0.05 were considered as differentially expressed between high- and low-FCR chickens. Then, the expression levels of differentially expressed probes were averaged by gene.

In total, 50 genes with the lowest raw *p*-values among 11,829 expressed genes were selected. The scaled expression levels of these genes were visualized in a cluster heatmap to identify clusters of genes and clusters of samples with similar profiles. A gradient color from green, which corresponded to the lowest expression, to red, which corresponded to the highest expression, was used in the heatmap. Hierarchical clustering of the scaled gene expression levels of top genes was based on the Pearson’s correlation distance and a Ward aggregation criterion.

### 2.11. Co-Expression Network Analysis

To study the correlations between gene expression levels and quantitative phenotypes as a complementary approach to the differential analysis, we applied a co-expression analysis using the R package weighted correlation network analysis (WGCNA) [27].

Briefly, in the first step, the WGCNA package constructs the network of co-expressed genes by creating an adjacency matrix that groups correlated genes into modules based on the pairwise correlations that are calculated using the Pearson correlation. To keep the network consistent with scale-free topology, the pickSoftThreshold function is used to analyze the network topology and to choose an appropriate softthresholding power value (β) to build the network and to allow the mean connectivity of all of the genes in the module to be evaluated.

Network construction was performed using an expression matrix that was constructed using 21 samples and 11,829 expressed and annotated genes. The unsigned connected network was built using the adjacency matrix between genes. From the gene expression matrix, pairwise Pearson correlations between genes were computed and were raised to a selected power of β = 9 using the pickSoftThreshold function to reach a scale-free topology index (R²) of at least 0.80, which fits our data the best. The adjacency matrix was organized into a topological overlap measure matrix, which assesses the degree of shared neighbors between pairs of genes.

### 2.12. Module Identification

A hierarchical clustering of the genes based on the topological overlap dissimilarity measure and average link was performed followed by a modular height cut-off value for branches in the hierarchical tree using the cutreeDynamic function (deepSplit = 2, minClusterSize = 30) to detect modules of co-expressed genes. The eigengene, which was the first principal component of the module and represented the expression values, was calculated for each module. Modules with expression profiles that were very similar were merged (height cut-off of 0.15 in the dendrogram) because there was a good probability that the genes belonging to these modules were highly co-expressed.

### 2.13. Module-Trait Relationships

The eigengene module was used to detect biologically relevant modules. Module–trait relationships were estimated using Pearson’s correlations (*p* < 0.05) between the eigengene module and each trait of interest (FE and meat quality parameters). Genes for each module with high significance levels (≥0.6) and that corresponded to the absolute value of the correlation between gene expression and the trait of interest and high module membership values (≥0.7) corresponding to the absolute value of the correlation of the module eigengene and the gene expression profile were defined as hub genes.

### 2.14. Functional Gene Ontology (GO) Enrichment Analysis

All of the expressed genes on the chip were annotated using GO [28,29] for Biological Process categories in the NCBI EntrezGene database using orthologs. Functional enrichment tests were performed using the ViSEAGO R package [30], which is available at http://bioconductor.org/packages/ViSEAGO/, accessed on 11 October 2021. A functional enrichment, supported by GO terms, within each module of genes from the WGCNA analysis was determined using Fisher’s exact tests and the “classic” algorithm (*p* < 0.01), with all of the the expressed genes used as the background. Biological functions were explored using the concept of semantic similarity developed by Wang et al. [31].

## 3. Results and Discussion

### 3.1. Phenotypic Characterization of KR Chickens

Descriptive statistics for FE as well as meat composition and quality are reported in Table 1. As expected, when the chickens were separated into two groups based on FCR or RFI, FE significantly differed between the groups. Variations in FCR were associated with significant changes in the biochemical composition of the thigh meat, which showed an increase in the lipid and AMP contents in the LFCR birds but a decrease in protein (amide I). In contrast, no significant variations in the physicochemical characteristics (ultimate pH, WHC, and DL) or in the carbohydrate and glycogen contents were observed. Abasht et al. [15] highlighted metabolic differences in the breast muscle between fast-growing chickens with high and low FE values. They showed a downregulation of glycolytic metabolism and lower glycogen content in the high-FE birds, whereas lipid metabolism (including lipid uptake and cholesterol synthesis) was upregulated. Here, the increased lipid content in the thigh muscles of the LFCR chickens was consistent with the latter observation by Abasht et al. [15]; however, whether similar mechanisms are involved in the two types of muscles remains unknown. In contrast, we did not observe any impact on carbohydrate metabolism in the current study, which is most likely because breast and thigh muscles have different metabolic profiles (fully glycolytic or more oxidative) and perhaps because the birds used in the studies had different genetic backgrounds.

### 3.2. Differential Analysis of Gene Expression

To compare the gene expression profiles in the thigh muscles between the HFCR and LFCR groups, a custom Agilent chicken 8 × 60 K Gene Expression Microarray was used for transcriptomic analysis. Among the 61,657 probes spotted on the array, 58,697 (95%) were expressed. Unfortunately, no genes were identified as being significantly differentially expressed between the two FCR groups. The same result was found between the RFI groups. However, the cluster heatmap of the 50 genes with the lowest raw *p*-values in the differential analysis (Figure 1) suggested a gene signature that discriminated the high- and low-FCR groups. Therefore, a WGCNA was performed to establish gene modules with similar expression profiles and to determine their correlations with the quantitative traits related to FE and meat quality. For distinction, each module is represented by a specific color name.

### 3.3. WGCNA and Module Identification

A gene network approach using WGCNA analysis is effective in combining different high-throughput transcriptome profiling and feed efficiency, meat quality data to pinpoint key functional gene modules, and molecular signaling. The hierarchical clustering of 11,829 expressed and annotated genes revealed 52 distinct co-expression modules. Some modules that shared neighbors between pairs of genes were then grouped together. In total, 38 modules (representing 10,562 genes) were kept. The remaining modules, which included 32 to 1886 genes, were investigated (Table 2).

### 3.4. Correlations between Phenotypic Data and Gene Modules in Thigh Muscle

Correlations between the 38 modules and recorded phenotypic traits, such as FE (FCR, and RFI), and meat quality traits, such as flavor, technological qualities, and biochemical compounds, are illustrated in Figure 2. For FE, modules MEmagenta (*p* = 0.02) and MEthistle1 (*p* = 0.02) were significantly correlated with FCR. Additionally, the modules MElightcyan (*p* = 0.04), MEdarkmagenta (*p* = 0.006), MEgrey60 (*p* = 0.006), MEdarkslateblue (*p* = 0.05), and MEdarkolivegreen (*p* = 0.02) were significantly correlated with RFI. Moreover, the modules MEskyblue, MEsteelblue, MEmediumpurple3, MEthistle2, and MEred were correlated with FCR and/or RFI FE traits as well as with several meat quality parameters (WHC, DL, IMP, AMP, and inosine). Furthermore, 21 hub genes, which were highly correlated with traits (gene significance ≥ 0.6) and the eigengene module (module membership ≥ 0.7), were identified. Among the genes belonging to the MEmediumpurple3 module, four hub genes were identified: the assembly competence domain (*ACD*), baculoviral IAP repeat containing 5 (*BIRC5*), cytochrome c oxidase assembly factor 3 (*COA3*), and myosin light chain 9 (*MYL9*). These genes may be involved in FE and thigh-meat quality.

The *ACD* gene is a conserved 29 residue sequence at the C-terminus of the sarcomeric myosin rod domain (coiled-coil rod) that is required for the tails of myosin dimers to self-assemble into antiparallel arrays in the second step of myosin heavy chain type II’s assembly into filaments [32,33]. Ikebe et al. [34] reported that different *ACD* regions are important for the filament formation of smooth muscle and non-muscle myosin II. The *BIRC5* gene is a target gene of miR-133a (muscle-related miRNA in mammals), which is associated with skeletal muscle development in chicken breasts and thighs [35]. According to Zhu et al. [36], *BIRC**5* expression is enriched during cell division, chromosomal segregation, and inflammatory responses in mice. Additionally, *BIRC5* might be critical in modulating the effects of microbiota on intestinal health. Neufert et al. [37] showed that *BIRC**5* limits bacterial growth, thereby contributing to mucosal wound healing. The *COA3* gene encodes a mitochondrial membrane protein that is required for the negative feedback regulation of cytochrome c oxidase 1 (*COX1*) translation in mitochondria. Both COA1 and COA3 link with Shy1 to form an early assembly intermediate of *COX**1* [38,39]. Additionally, the lack of *COA3* function traps Mss51 in the committed state and promotes *COX**1* synthesis. In humans, *COA3* stabilizes *COX**1* and supports its assembly with COX-partner subunits, and this is associated with mitochondrial diseases [40]. The *MYL**9* gene encodes a regulatory light chain that is involved in the stabilization of myosin II and cellular integrity. It interacts with a variety of non-muscle types of myosin heavy chain type II [41]. In chickens, *MYL**9* is associated with muscle-fiber development [42]. In mice, a *MYL**9* gene knockout resulted in a gastrointestinal motility disorder [43]. Because of their functions, we hypothesized that these genes affect both FE and meat quality.

### 3.5. Functional Enrichment Significance of Network Modules 

Functional analyses of the modules correlated with either FCR only (MEmagenta and Methistle1), RFI only (Melightcyan, Medarkmagenta, Megrey60, Medarkslateblue, and Medarkolivegreen), or FCR, RFI, and meat quality parameters (MEskyblue, MEsteelblue, MEmediumpurple3, MEthistle2, and MEred) included 730, 775, and 316 genes, respectively, and highlighted 169, 61, and 61 enriched GO terms, respectively. Information on significant GO terms is reported in Appendix A.

A summary of the numbers of genes and GO terms related to FE and thigh-meat quality is presented in Table 3. For FCR only, the enriched GO terms were related to the immune system (77 GO terms), cell activation (27), biological process (27), metabolic process (21), cell locomotion (12), tissue maintenance (3), and nucleotide metabolism (2). The GO terms related to RFI only were cell activation (17), nucleotide metabolism (17), immune system (7), biological process (5), transport process (4), organ development (4), skeletal organization (4), and metabolic process (3). Additionally, genes correlated with FCR, RFI, and meat quality were enriched in 61 GO terms, as follows: nucleotide metabolism (17), cell activation (17), immune system (7), biological process (5), organ development (4), transport process (4), skeletal muscle organization (4), and metabolic process (3).

The main aims of this study were to determine whether a change in FE impacted meat quality and to investigate any related molecular mechanisms. The study results indicated that FE did affect meat quality and, using a WGCNA and functional enrichment analysis, key molecular pathways related to the FE and meat quality (especially WHC, DL, IMP, AMP, and inosine) of thigh muscles of KR chickens were revealed. Alterations in the regulation of biological and metabolic processes (such as nucleotide metabolism, fatty acid metabolic process, and oxidative stress) may explain why genes related to metabolic processes, such as nucleotide metabolism, lipid uptake and transport, lipid catabolism, and cholesterol synthesis, were differentially expressed in breast muscles between high- and low-FE chickens [15]. Additionally, Petracci et al. [44] indicated that oxidative stress induces a change in connective tissue synthesis (collagen, proteoglycan, and glycosaminoglycan). Furthermore, Yang et al. [18] showed that low-RFI chickens synthesize ATP more efficiently and control reactive oxygen species production more strictly in breast muscles by enhancing the mitochondrial functions in skeletal muscles compared to in high-RFI chickens. Thus, a change in FE affects meat quality parameters through oxidative stress, nucleotide metabolic, and biological process pathways in both breast and thigh muscles. Interestingly, variations in FCR-affected pathways are related to immunity and adaptive immune responses, such as the regulation of leukocyte activation, lymphocyte activation, T-cell activation, and type I interferon production, whereas variations in RFI had greater impacts on nucleotide metabolism and cell activation (i.e., cellular protein-containing complex assembly and the regulation of cellular catabolic process). Thus, this study suggested that selection based on FE may also influence immune responses and flavor precursors in chickens.

## Figures and Tables

**Figure 1 animals-11-02977-f001:**
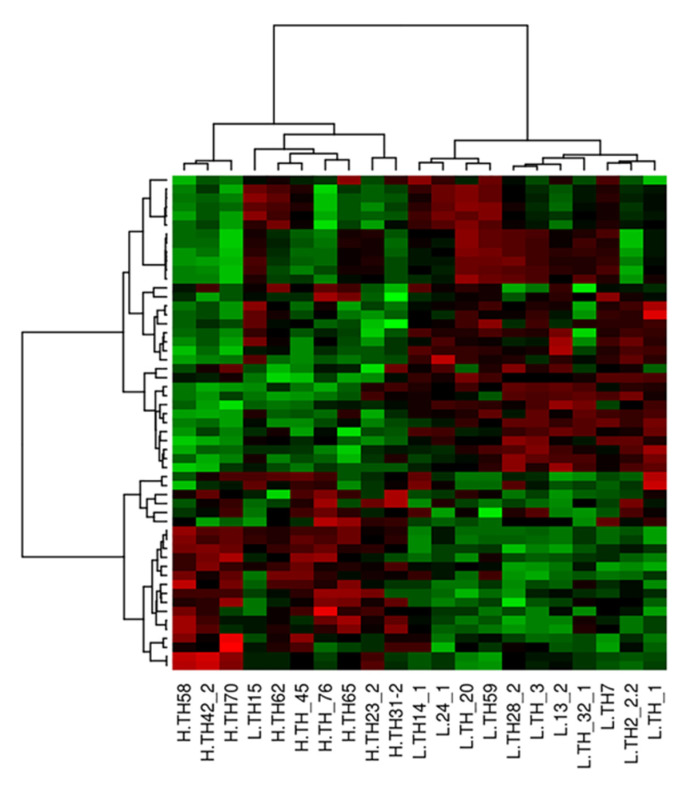
Hierarchical clustering of the top 50 genes (lowest *p*-values) between high- and low- feed conversion ratio (FCR) in Korat chickens. Samples are shown in columns, and genes are shown in rows. The scaled expression levels are depicted using a color gradient: upregulated and downregulated genes are shown in red and green, respectively. Genes and samples were grouped using hierarchical clustering analyses. The hierarchical clustering of the scaled gene expression matrix was based on Pearson’s correlations and average link aggregation distances.

**Figure 2 animals-11-02977-f002:**
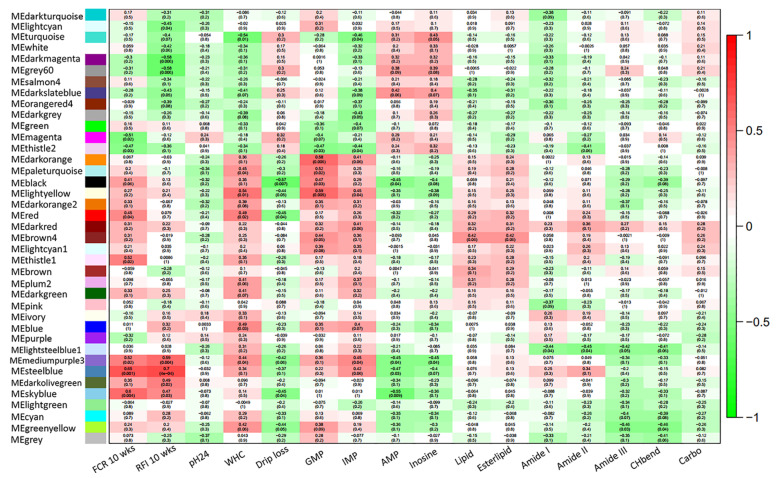
Correlations between modules and feed efficiency, meat quality traits, or both in the thigh muscles of Korat chickens. Each row corresponds to a module, and each column corresponds to a trait. Each cell contains the correlation coefficient and the *p*-value in the first and second lines, respectively. The table is color-coded by correlation in accordance with the legend. The module name is shown on the left side of each cell. FCR10 wks; feed conversion ratio at 10 weeks of age; RFI10 wks, residual feed intake at 10 weeks of age; WHC, water-holding capacity; GMP, guanosine monophosphate; IMP, inosine monophosphate; AMP, adenosine monophosphate; Esterlipid, ester carbonyl of phospholipids; CHbend, CH bending; and Carbo, carbohydrate and glycogen.

**Table 1 animals-11-02977-t001:** Comparison of meat quality parameters in thigh muscles between low- and high-feed efficiency Korat chickens (means ± SE).

**Feed Efficiency Parameters**	**Groups**		***p*-Values**
	**Low**	**High**	
FCR	2.12 ± 0.39	3.07 ± 0.30	0.00 **
RFI	−588.22 ± 43.89	671.11 ± 36.16	0.00 **
**Meat quality parameters**	**Groups**		
	**LFCR**	**HFCR**	
Ultimate pH	6.06 ± 0.07	6.14 ± 0.10	0.50
WHC (%)	81.10 ± 1.11	83.45 ± 2.00	0.29
DL (%)	11.16 ± 1.24	8.30 ± 1.24	0.14
GMP (mg/g)	0.15 ± 0.01	0.14 ± 0.01	0.57
IMP (mg/g)	4.25 ± 0.33	4.57 ± 0.27	0.49
AMP (mg/g)	0.11 ± 0.01	0.08 ± 0.00	0.02 *
Inosine (mg/g)	0.50 ± 0.05	0.40 ± 0.03	0.13
Lipid (%)	22.02 ± 1.40	16.15 ± 1.29	0.02 *
Ester carbonyl of phospholipids (%)	6.13 ± 1.26	5.64 ± 1.22	0.78
Amide I (%)	22.16 ± 0.83	25.96 ± 1.19	0.03 *
Amide II (%)	17.18 ± 0.68	18.92 ± 0.55	0.07
CH bending (%)	16.36 ± 0.70	17.03 ± 0.70	0.51
Amide III (%)	0.56 ± 0.05	0.55 ± 0.04	0.91
Carbohydrate and glycogen (%)	15.28 ± 0.56	14.49 ± 0.59	0.35

* *p* ≤ 0.05 and ** *p* ≤ 0.01. Abbreviations: FCR, feed conversion ratio; RFI, residual feed intake; LFCR, low FCR; HFCR, high FCR; WHC, water-holding capacity; DL, drip loss; GMP, guanosine monophosphate; IMP, inosine monophosphate; AMP, adenosine monophosphate.

**Table 2 animals-11-02977-t002:** Numbers of genes in modules from a weighted gene co-expression network analysis of Korat chickens.

Modules	Numbers of Genes	Modules	Numbers of Genes	Modules	Numbers of Genes
MEdarkturquoise	277	MEthistle2	53	MEpink	860
MElightcyan	164	MEdarkorange	122	MEivory	73
MEwhite	121	MEpaleturquoise	114	MEblue	907
MEdarkmagenta	289	MEblack	329	MEpurple	266
MEgrey60	152	MElightyellow	151	MElightsteelblue1	79
MEsalmon4	32	MEdarkorange2	66	MEmediumpurple3	82
MEdarkslateblue	58	MEred	340	MEsteelblue	116
MEorangered4	190	MEdarkred	148	MEdarkolivergreen	112
MEdarkgrey	718	MEbrown4	62	MEskyblue	118
MEgreen	681	MEplum2	166	MElightgreen	152
MEmagenta	289	MEdarkgreen	145	MEcyan	207
MEgreenyellow	238	MEgrey	1267	MEturquoise	1886
MElightcyan1	73	MEthistle1	48	MEbrown	855

**Table 3 animals-11-02977-t003:** Summary of the numbers of genes and gene ontology (GO) terms involved in the feed efficiency and thigh-meat quality of Korat chickens.

Traits	Modules	Numbers of Genes	Numbers of Enriched GO Terms	Main Enriched GO Terms
FCR	MEmagenta, MEthistle1, MEthistle2, MEred	730	169	immune system, cell activation, biological process, metabolic process, cell locomotion, tissue maintenance, nucleotide metabolism
RFI	MElightcyan, MEdarkmagenta, MEgrey60, MEdarkslateblue, MEdarkolivegreen,	775	61	cell activation, nucleotide metabolism, immune system, biological process, transport process, organ development, skeletal organization, metabolic process
FCR, RFI and meat quality	MEskyblue, MEsteelblue, MEmediumpurple3	316	61	biological process, cell activation, immune response, metabolic process, nucleotide metabolism, organ development, skeletal muscle organization, transport process

Abbreviations: FCR, feed conversion ratio; RFI, residual feed intake.

## Data Availability

Raw sequence data are available from the NCBI (https://www.ncbi.nlm.nih.gov/geo/query/acc.cgi?acc=GSE162848, accessed on 11 October 2021), accession number GSE162848.

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
