# Peer review of "Revealing Pathways Associated with Feed Efficiency and Meat Quality Traits in Slow-Growing Chickens"

_animals, 2021, doi:10.3390/ani11102977_

Round 1

Reviewer 1 Report

The authors of "Revealing Pathways Associated with Feed Efficiency and Meat Quality Traits in Slow-Growing Chickens" performed a WGCNA using a micro-array data in order to investigate putative co-expressed gene networkds associated with FE and meat quality traits. However, there is extremely important issue regarding the WGCNA analysis performed in teh current study. The authors mentioned in the introduction that "the aim of this study was to investigate genes and molecular pathways involved in the FE and thigh-meat quality of slow-growing KR chickens". However, the co-expressed gene networks were estimated using all the samples simultaneously in the same expression matrix. If the authors want to identify biological processes associated with the feed efficiency status and meet quality, a different experimental design should be applied. The authors could use the WGCNA package to compute the consensus co-expressed modules (as performed in the manuscript) and the co-expressed modules for each group (High-FCR and low-FCR). After this procedure, the overlapping between groups of the consensus modules and the modules observed in each group should be caculated. Subsequently, for example, those modules observed in the High-FCR with no counterpart in the consensus modules could be considered as modules exclusively co-expressed in this group of samples (the same should be applied to the low-FCR group). Finally, the correlation between the meat quality traits and the exclusively co-expressed gene networks should be calculated. The main theory behind a gene co-expressed analysis is that the genes which play similar biological roles will have the expression modulated in a similar way. Consequently, alterations in the enviroment or differences in contrasting groups will be reflected in the co-expression profile. Therefore, if the authors want to describe biological processes that are associated with different FE groups and meat quality properties only the exclusively co-expressed gene networks should be taking into account. There different approaches to estimate the exlcusively co-expressed gene networks, for example, comparing directly the overlapping between the modules identified in each group (not computing the consensus modules). Ths information is available in the WGCNA documentation.

Minor comments:

Lines 128-130: Cite mean and SD for the whole dataset.

Line 204: Corresponding originally to how many trancripts?

Line 230: "Differential expression analysis"

Line 231: Cite the reference for limma. It is a very commonly used package. However, the citation is still required.

Line 242: Why 50? Is there a reference for this number? How the samples are clustered using all the genes?

Line 284: Adjusted p-value? If not, the multiple testing correction should be applied.

Figure 2: Impossible to see the values.

Reviewer 2 Report

Main suggestions

-Although in the context of this paper FE can mean both FCR and RFI, the authers should be consistant with the use of these terms.

- On line 187 the authors have written that "Chickens with the 9 highest and 12 lowest FE values at 10 weeks of age were selected", also on line 317 the authors mentioned that "The same 
result was found between RFI groups." However, in the Materials and Methods section the grouping of birds was only described based on FCR. 

- To avoid possible confusion the authors can perhaps mention in the Materials and Methods section that the division of birds into two groups can be based on either FCR or RFI.  

- The authors should mention in the begining of Section 2.5 "Fourier-transform Infrared Spectroscopy (FTIR) Analysis" the purpose of performing this analysis.

- The description given in Sections 2.5 and 2.6 belong to the same analysis. Section 2.6 can be made either a subsection of Section 2.5 or can directly be added to Section 2.5.  

- Sections 2.12, 2.13, 2.14 and 2.15 explain different segments of the same analysis, which is weighted correlation network analysis (WGCNA). Making separate sections for the same analysis 
pipeline make it difficult to understand. Make sections 2.13, 2.14, and 2.15 subsections of the Section 2.12.

- In the WGCNA different values have been assigned to different parameters or used as threshold, for example, "scale-free topology 258 index (R²) of at least 0.80." have been mentioned in 
line 259 of Section 2.13. No reason for choosing these particular values has been provided. Perhaps some references can be provided in the explaination of this analysis that can 
justify the use of these values. 

- Overall the explaination of WGCNA analysis needs to be improved to make the flow of information from one segment of the analysis to another more comprehensible.
For example, in lines 257-258 "raised to a selected power of β = 9 using the pickSoftThreshold function".
Here the symbol β does not have any meaning. Such issues have to be resolved in the 
explaination of this analysis.

Some minor suggestions: 

Introduction:

-The following sentence in lines 46-47 should be rewritten: 

"These criteria have improved over the years in fast-growing production systems through targeted selection and a reduction in the slaughter age [5]."

as reduction in the slaughter age must also be in response to selection. Clearify the sentence.

- Line 83 needs correction: 

"Leg muscles (thigh and drumstick), however, are preferred to thigh muscles in East Asia, Mexico, India, Russia, and Morocco [19]"

I think the authors wanted to say that leg muscles (thigh and drumstick), however, are preferred to "breast" muscles.

Rasults and discussion:

- In line 332 it has been mentioned that "Some modules were then grouped together.",  but no reason has been provided. Although this information has been given in the Materials and Methods 
section, it is better to provide only the number of modules after grouping or give the reason of this grouping again at this point.   

- Figure 1: between high- and low- feed 324 conversion ratio (FCR) Korat chickens. Put "in" before "Korat chicken". 

- Add "s" to the header of the fifth column in  table 2 to make it "Modules".

- Mention somewhere that the output the software WGCNA represents the different gene modules in the form of colors and that these color names do not have anything to do with gene names 
or functions.

- On line 408, in the sentence "The study results indicated that Fe did affect meat quality", "E" should be capital.

- Jutify the use of WGCNA in the discussion section.

- The sentence "pathways related to FE and meat quality (especially WHC, DL, IMP, AMP, and inosine) in thigh muscles of KR chickens." Replaced "in thigh" by "of thigh".

Reviewer 3 Report

In this study, no differentially expressed genes were identified between the two FCR groups in Slow-Growing Chickens (Korat). And a weighted gene co-expression network analysis (WGCNA) was further performed to determine the correlations between co-expressed gene modules and FE, thigh-meat quality, or both. 12 modules and several molecular factors were identified which might be correlated with FE and some meat quality traits. This research is meaningful and can be accepted after revision.

  1. The title of this manuscript was “Revealing Pathways Associated with Feed Efficiency and Meat Quality Traits”. In the article, authors also mentioned that the aim of this study was to investigate genes and molecular pathways involved in the FE and thigh-meat quality of slow-growing KR chickens. However, I only found the authors revealed the modules and several molecular factors were associated with FE and meat quality. I did not find the pathways the authors revealed in this research.
  2. Line 50-51. The authors showed that “Slow-growing chickens are slaughtered later than fast-growing chickens, mainly at approximately 10 weeks of age”. I did not agree with the statement slow-growing chickens are slaughtered mainly at approximately 10 weeks of age.
  3. Line 56. “Korat (KR) is a new alternative meat-type chicken.” What the authors want to express using the word “alternative”. Besides, what is the body weight of this chicken breed at 10 weeks of age?
  4. Line 128-129. I want to know what is the body weight of the two groups, and whether their existed difference between them.
  5. Line 332-334. Authors mentioned that 38 modules were kept and the gray module was eliminated from further analyses. In line 339, the author stated that “Correlations between the 38 modules and recorded phenotypic traits,….”. While in Figure 2,39 modules were showed.

Round 2

Reviewer 2 Report

The corrections are fine. Only the following points should be considered:

- Page 6 line 271: Change the line "at least 0.80 that to fit best to our data." to "at least 0.80 which fits our data best."

- Page 10 line 376: The correction made by the authors is not correct.
My suggestion was to "Mention somewhere that the output the software WGCNA represents the different gene modules in the form of colors and that these color names do not have anything to do with gene names or functions." by which I meant that the color names mentioned as module names, for example MEgrey or MElightgreen. These names do not represent the legend color coding so stating "The table is color-coded by correlation in accordance with the legend which the color is not related with genes name or function." is not correct.

I suggest the following wording on line 332 instead: "For distinction, each module is represented by a specific color name." 
